

# Comprehensive analysis of particle growth rates from nucleation mode to cloud condensation nuclei in Boreal forest

Pauli Paasonen[1], Maija Peltola[1], Jenni Kontkanen[1], Heikki Junninen[1,2], Veli-Matti Kerminen[1], Markku Kulmala[1,3,4]

[1]Institute for Atmospheric and Earth System Research (INAR) /Physics, Faculty of Science, University of Helsinki, Finland
[2]Institute of Physics, University of Tartu, Ülikooli 18, EE-50090 Tartu, Estonia
[3]Aerosol and Haze Laboratory, Beijing Advanced Innovation Center for Soft Matter Science and Engineering, Beijing University of Chemical Technology, 100029 Beijing, P.R. China
[4]Joint International Research Laboratory of Atmospheric and Earth System Sciences, School of Atmospheric Sciences,
Nanjing University, 210046 Nanjing, P.R. China

*Correspondence to*: Pauli Paasonen (pauli.paasonen@helsinki.fi)

**Abstract.** Growth of aerosol particles to sizes at which they can act as cloud condensation nuclei (CCN) is a crucial factor in estimating the current and future impacts of aerosol cloud climate interactions. Growth rates are typically determined for
particles with diameters ($d_P$) smaller than 40 nm immediately after a regional new particle formation (NPF) event. These growth rates are often taken as representatives for the particle growth until CCN sizes ($d_P > 50$-$100$ nm). In modelling frameworks, the concentration of the condensable vapours causing the growth is typically calculated with steady state assumptions, where the condensation sink is the only loss term for the vapours. Additionally, the growth to CCN sizes is represented with the condensation of extremely low-volatile vapours and gas-particle partitioning of semi-volatile vapours.
Here, we use a novel automatic method to determine growth rates (GR) from below 10 nm to hundreds of nanometres from a 20-years long particle size distribution data set in Boreal forest. With this method, we are able to detect growth rates also at other times than immediately after a NPF event. We show that the GR increases with an increasing oxidation rate of monoterpenes, which is closely coupled with the ambient temperature. Based on our analysis, the oxidation reactions of monoterpenes with ozone, hydroxyl radical and nitrate radical all are capable of producing vapours that contribute to the
particle growth in the studied size ranges. We find that GR increases with particle diameter, resulting in up to three-fold GRs for particles with $d_P \sim 100$ nm in comparison to those with $d_P \sim 10$ nm. We use a single particle model to show that this increase in GR can be explained with aerosol-phase reactions, in which semi-volatile vapours form non-volatile dimers. Finally, our analysis reveals that the GR of particles with $d_P < 100$ nm is not limited by the condensation sink, even though the GR of larger particles is. Our findings suggest that in the Boreal continental environment, the formation of CCN from
NPF or sub-100 nm emissions is more effective than previously thought, and that the formation of CCN is not as strongly self-limiting process as the previous estimates have suggested.



## 1 Introduction

The role of aerosol particles in global climate is one of the largest uncertainties in our current knowledge of the climate system (Boucher et al., 2013). Aerosol particles that are large enough, having diameters ($d_P$) larger than about 50-100 nm, can act as cloud condensation nuclei (CCN) which are cores of all the cloud droplets in our atmosphere (Kerminen et al. 2012). Since the lifetime and albedo of a cloud depend on the CCN concentration, they significantly adjust the fraction of solar radiation reflected back to space (Boucher et al., 2013). Cloud condensation nuclei are emitted to the atmosphere directly from both anthropogenic (Paasonen et al., 2016) and biogenic sources (Després et al., 2012), but a significant fraction of CCN are formed by the condensation growth of smaller particles (Merikanto et al., 2009; Kerminen et al., 2012; Paasonen et al., 2013; Dunne et al., 2016; Gordon et al., 2017). These smaller particles may originate from atmospheric new particle formation (NPF), anthropogenic combustion or other emissions. The importance of the growth due to condensation of biogenic vapours has been shown both in model and observational studies (Merikanto et al., 2009; Makkonen et al., 2012; Paasonen et al., 2013; Scott et al., 2014), where the fraction of CCN originating from the growth of smaller particles is estimated to be around 50 % of the total CCN concentration.

The condensable biogenic vapours typically originate from emissions of volatile organic compounds (VOC) from plants (e.g. Kulmala et al., 1998; Riipinen et al., 2011). In the atmosphere, VOCs are oxidized mainly by ozone ($O_3$), hydroxyl radical (OH) and nitrate radical ($NO_3$), which decreases their volatility (e.g. Atkinson and Arey, 2003; Kroll and Seinfeld, 2008). One oxidation step cannot decrease the volatility enough for allowing the vapour to condense, but the required number of oxygen molecules in an extremely low volatile organic compound is roughly the same as the number of carbon molecules (Ehn et al., 2014; Jokinen et al., 2015). However, an auto-oxidation process in which, after the initial oxidation, further oxidation steps occurs with atmospheric oxygen molecules can produce very rapidly condensable vapours from VOCs (Ehn et al., 2014; Barsanti et al., 2017). In addition to extremely low volatile organic compounds, low- or semi-volatile organic compounds can participate in aerosol growth by moving towards equilibrium in gas-particle partitioning. The impact of different volatility vapours is often analysed using so called Volatility Basis Set (VBS, Donahue et al., 2011), in which the compounds with roughly similar volatilities are lumped together in order to facilitate e.g. modelling their impact on the growth rate.

The growth rate of atmospheric particles can be determined after a period during which formation of particles with roughly similar size has occurred on a regional scale. Typically, the growth rates are determined after atmospheric NPF events, during which new particles are simultaneously formed from vapour molecules in a large area (Kulmala et al., 2012). After a NPF event, the growth of the formed particle mode can be typically followed up to 15 nm or sometimes up to 50 nm, but very rarely up to 100 nm. In order to observe the growth until 100 nm at the measurement station under typical conditions, simultaneous NPF should happen in a very large area (e.g. with wind speed 5 m/s and growth rate of 3 nm/h, from the station to roughly 600 km upwind from the station), followed by continuous rather homogenous conditions without disturbing major aerosol sources. Since these kinds of circumstances are encountered only in specific clean environments and even in them





only under suitable conditions, the growth rates observed from NPF to 100 nm cannot be considered as representative for wide spatial and temporal scales.

The growth rate has been shown to increase with increasing particle diameter in nucleation mode ($d_p < 25$ nm) (Manninen et al., 2010; Yli-Juuti et al., 2011; Kuang et al., 2012; Häkkinen et al., 2013; Dos Santos et al., 2015). This is assumed to be caused by the decreasing impact of the Kelvin effect, which makes condensation more difficult over surfaces with a strong curvature. Recently, a case study in remote Arctic environment suggested that the particle growth rate is higher in the Aitken mode ($25$ nm $< d_P < 100$ nm) than in the nucleation mode, which follows from the growth caused by partitioning of semi-volatile vapours (Burkart et al., 2018). In these size ranges, the increase in particle growth rate with diameter is suggested to result from particle-phase reactions, e.g. dimerization of the semi-volatile vapours (Apsokardu and Johnston, 2018). However, it was also suggested that the increasing viscosity of particles with increasing size would slow down the growth rate (Zaveri et al., 2017).

Here, we first present an easy-to-use automatic method to determine the particle growth rates from particle number size distribution data by analysing growing particle modes that do not need to immediately follow the NPF event. The growth rates can be calculated for different particle size ranges: nucleation mode ($d_P < 25$ nm), Aitken mode ($25$ nm $< d_P < 100$ nm) and accumulation mode ($d_P > 100$ nm). The method is based on the manual growth rate analysis presented in Arneth et al. (2016). Secondly, we determine the growth rates at the SMEAR II station in Hyytiälä, Finland, in different seasons and times of day during 20 years. Finally, we determine the impacts of atmospheric conditions, estimated sources and sinks of condensable biogenic vapours and particle diameter on the growth rate. With the new method we have, for the first time, comprehensive enough data set for a detailed analysis of the growth in all the sizes from the nucleation mode to CCN sizes and beyond.

## 2 Materials and methods

### 2.1 DMPS data set and other applied variables

The automatic determination of growth rates, described in more detail in Section 2.2., was developed using the particle size distribution (PSD) data recorded at the SMEAR II station (Hari and Kulmala, 2005) with a Differential Mobility Particle Sizer (DMPS, Aalto et al., 2001) system. The applied data set is 20 years long, from January 1996 to August 2016, and presents the PSD for particles in diameter range from 3 to 1000 nm. The measurement station is situated in a boreal forest area with dominant tree species being Scots pine (*Pinus Sylvestris*). The closest densely-habituated area is the city of Tampere, roughly 80 km west from the station.

The determined growth rates were compared with meteorological variables and gaseous compounds, such as ozone, sulphur dioxide and nitrogen oxides, recorded at SMEAR II. The temperature was measured with PT-100 sensor at 16.8 m height



and the ozone concentration with ozone analyser (TEI 49C, Thermo Fisher Scientific, Waltham, MA, USA). These data (as well as the data for several other parameters, which were investigated in terms of their connection to growth rates but which we do not present in this manuscript) together with more detailed explanation of their measurements can be found in the AVAA database (http://avaa.tdata.fi/web/smart/). The condensation sink (CS), describing the loss rate of condensable

vapours due to their condensation onto aerosol particles, was calculated from the PSD data using the methods described in Kulmala et al. (2001).

Next, the growth rates were compared with monoterpene concentrations ([MT]) and related parameters determined using proxies developed by Kontkanen et al. (2016). The applied proxy for monoterpene concentrations is given in Eq. (12) in Kontkanen et al. (2016). They showed that the correlation coefficient between this proxy and measured concentration was

0.74, and that for 80 % of the data points the proxy had a bias smaller than a factor of 5.8, which is rather small considering that the monoterpene concentration varies over almost three orders of magnitude. In addition to [MT], we inspected the correlation of GR with the proxy of monoterpene oxidation products

$$[MT_{Ox}] = \frac{[MT] \times (k_{MT+OH}[OH] + k_{MT+O_3}[O_3] + k_{MT+NO_3}[NO_3])}{CS},$$    (1)

where $k_{MT+X}$ is the reaction rate coefficient for α-pinene and oxidant $X$, [OH] is calculated with a radiation-based proxy

generated for Hyytiälä by Petäjä et al. (2009) and [NO$_3$] is calculated based on Peräkylä et al. (2014) as described in Kontkanen et al. (2016), where also the other details and assumptions for the proxies are described.

Finally, we compared the GRs with the source rate of monoterpene oxidation products i.e. the oxidation rate of monoterpenes (OxRate), which is the nominator in the right-hand side of Eq. (1). , and tested different weighting factors for the different terms (different oxidants) therein. Because the different oxidation reactions are expected to have different yields

of semi-volatile compounds (Jokinen et al., 2015), we tested whether introducing separate weighting factors, varied from 0.01 to 100, for OH and NO$_3$ oxidation reactions in Eq (1) would improve the correlation between the oxidation rate of monoterpenes and GR. The weighting factors were optimized by minimizing the inverse of the Pearson's correlation coefficient ($1/r$) with the Matlab function *fminsearch*, and the initial conditions were varied in order to confirm that the results do not represent only local minima. It is to be noted that, because the optimization concerned only the relative shares

of different oxidation reactions and $r$ is not sensitive to the absolute values of the data points, setting the weighting factor for ozonolysis reaction to 1 does not impact the results.

## 2.2 Automatic method for determining the growth rate

The DMPS data, described in Sect. 2.1., is first smoothed over five time steps with a median filter. Peak diameters (marked as white circles in Fig. 1) are determined from the smoothed data for each size distribution by fitting parabola to logarithmic

particle concentrations in size bins around local concentration maxima. The growth rates are determined by making linear least squares fits to these peak diameters as a function of time, if they fulfil the below described criteria. In the following



description the PSDs are marked with PSD$_n$ so that for the first PSD determined for the day $n=0$ and for the next $n=1$ and so on.

The peaks of PSD$_{n\geq0}$ are divided to consecutive groups based on the time and diameter difference between them. If the first peak is determined in PSD$_0$, the timewise closest peak in PSD$_{n>0}$ is added to the same group, if it takes place within an hour

from PSD$_0$ and is close enough in size (maximum allowed difference is 10 nm for peaks with $d_P < 50$ nm and 50 nm for peaks with $d_P > 150$ nm). If this peak is found e.g. in PSD$_2$, the size distributions PSD$_{n>2}$ are inspected in a similar manner. The procedure is repeated and the group of peaks is extended as long as more points are found. The peaks falling out of the size (or temporal) range are inspected later similarly in order to see if they form a group with other peaks.

When all the peaks within the PSD data file (typically for one day) have been assigned to a group (which in some cases can

consist of only six points), the groups are inspected one by one in order to find periods with monotonic growth of the peak diameter within the groups. The monotonicity is determined with three conditions: *i*) temporal and diameter differences between consecutive peaks, *ii*) similarity of the growth rates, retrieved from linear least squares fits to the peaks, along the growth period, and *iii*) a combination of these two parameters. When these monotonicity conditions (described in more detail below) are violated for the third time, the growth period is ended. The peaks that cause the two first violations are excluded

from the growth period before continuing to the next PSD.

The maximum allowed temporal and diameter differences between consecutive peaks (condition *i* above) are 0.5 h and 20 nm, respectively, which are stricter limitations than when the grouping of the peaks is done. The condition for monotonicity (*ii* above) of the fitted growth rate is not fulfilled if both a) the addition of a new peak changes the growth rate by a factor larger than 1.5 in comparison to the growth rate during the first hour of the growth period, and b) the slope of the fit to the

peaks in the latest 3 PSDs differs by a factor larger than 2 from the growth rate during the first hour. The combined condition *iii* uses the original growth rate GR$_{orig}$ which is fitted for the first hour (or if the growth period is not yet one hour long, the growth rate of last 4 points), and the diameter of the new peak. The condition is fulfilled if the diameter of the new peak is between the diameters $1.5\times$GR$_{orig}\times$D$t+ b$ and GR$_{orig}/1.5\times$D$t + b$, where D$t$ is the time step between the last and the new peak and $b$ is a tolerance constant having the value of 10 % of the new peak diameter when $d_P > 20$ nm and 2 nm when $d_P < 20$

nm.

Finally, when the original growth periods of a minimum of one hour have been determined using the monotonicity conditions described above, each growth period is inspected to find out whether it can be combined with a previous or following growth period. This is done because growth periods shorter than 2 h are not considered long enough for determining the growth rate. Two growth periods are combined if their growth rates do not differ more than by a factor of

1.5 and if the growth rate of the combined growth period (retrieved from the linear least squares fit to the peaks included in both initial periods) does not differ more than by a factor of 1.5 from the former initial period. Additionally, the latter initial



period needs to start within a timeframe of at most half of the sum of the initial growth period durations, but not more than 2 hours, before and after the end of the former growth period.

In the analysis, the combined and non-combined growth periods are not separated. The minimum duration applied is 2 h, but in many parts of the Sect. 3 the results are also presented separately for periods with duration over 5 h.

**2.3 Model**

In order to investigate how the diameter of the particle, vapour concentration and particle phase chemistry affect the growth rate, we applied a simple one-particle process model. The model included a particle, which consists of extremely low volatile molecules (ELVOC), semi-volatile molecules (SVOC) and non-volatile dimers formed from SVOC in the particle phase (SVOC$_{dim}$). The parameters describing the model and vapours are shown in Table 1. The basic assumption of the model is

that the compounds are fully mixed within the particle. The model consists of a set of differential equations for the number of ELVOC and SVOC molecules and SVOC$_{dim}$ inside the particle, adopted from the theoretical frameworks by Fuchs and Sutugin (1970), Kerminen et al. (2000), Vesterinen et al. (2007) and Trump and Donahue (2014):

$$\frac{d[\text{ELVOC}]}{dt} = 2\pi D\beta d_P C_{\text{ELVOC}}, \tag{2}$$

$$\frac{d[\text{SVOC}]}{dt} = 2\pi D\beta d_P \left( C_{\text{SVOC}} - \text{Ke}\, C_{\text{SVOC,eq}} \right) - 2k_{\text{dim}} \left( \frac{[\text{SVOC}]}{V_P} \right)^2 V_P \tag{3}$$

and

$$\frac{d[\text{SVOC}_{\text{dim}}]}{dt} = 2k_{\text{dim}} \left( \frac{[\text{SVOC}]}{V_P} \right)^2 V_P. \tag{4}$$

Here [ELVOC], [SVOC] and [SVOC$_{dim}$] describe the number of ELVOC, SVOC and SVOC$_{dim}$ molecules in the particle, respectively, $D$ is vapour diffusion coefficient, $d_P$ is particle diameter, $C_{\text{ELVOC}}$ and $C_{\text{SVOC}}$ are the gas phase concentrations of ELVOC and SVOC, respectively, $k_{\text{dim}}$ is the reaction rate coefficient for the formation of SVOC dimers in the aerosol phase

and $V_P$ is the volume of the particle. In Eq. (2) it is assumed that $C_{\text{ELVOC}} \gg C_{\text{ELVOC,eq}}$. In Eqs. (2-3), $\beta$ describes the Fuchs-Sutugin correction factor for the transition regime

$$\beta = \frac{1 + \text{Kn}}{1 + 0.377\text{Kn} + 1.33\text{Kn}(1 + \text{Kn})}, \tag{5}$$

where Kn = 2×68 nm/$d_P$ is the Knudsen number. In Eq. (3), Ke is the Kelvin term

$$\text{Ke} = \exp\left( \frac{4\sigma V_m}{RT d_P} \right),$$

25        (6)

where $\sigma$ describes the surface tension, $V_m$ is the molar volume, $R$ is the ideal gas constant (8.314 kg m$^2$ s$^{-2}$ mol$^{-1}$ K$^{-1}$) and $T$ is the temperature. The equilibrium vapour concentration (Pankow, 1994) for gas phase SVOC is calculated as



$$C_{\text{SVOC,eq}} = C_{\text{SVOC,sat}} \frac{[\text{SVOC}]}{[\text{ELVOC}]+[\text{SVOC}]+[\text{SVOC}_{\text{dim}}]}, \qquad (7)$$

where $C_{\text{SVOC,sat}}$ is the saturation vapour concentration of SVOC, which is the inverse of the absorption partitioning coefficient in Kerminen et al. (2000).

The change in the diameter of the particle is calculated as

$$\frac{dd_P}{dt} = \frac{\left(\frac{d[\text{ELVOC}]}{dt}V_{\text{ELVOC}}+\frac{d[\text{SVOC}]}{dt}V_{\text{SVOC}}+\frac{d[\text{SVOC}_{\text{dim}}]}{dt}V_{\text{SVOC}_{\text{dim}}}\right)}{\frac{\pi}{2}d_P^2}, \qquad (8)$$

where $V_i = \frac{M_i}{N_A \rho}$ is molecular volume for compound $i$, calculated with compound molar mass $M_i$, Avogadro number $N_A$ ($6.022 \times 10^{23}$ # mol⁻¹) and density $\rho$.

The initial values for all the variables are given in Table 1.

## 3 Results and discussion

**3.1 Observed particle growth rates, seasonal and diurnal variations**

The number of determined growth rates (GR) in different size ranges during different times of the year and day are presented in Table 2. The number is the largest for the Aitken mode in summer and the smallest for the nucleation mode in winter.

The observed growth rates did not show a clear diurnal cycle (Fig. 2). This is rather surprising, since the strong diurnal cycles of oxidant concentrations, in terms of OH and nitrate radicals, would be expected to affect the concentrations of

condensable vapours and the growth rates. The possibility of the opposite diurnal cycles of these factors partly cancelling out their impact and further analysis on their effect is presented in Sect. 3.2.1.

In the nucleation and Aitken mode, the growth rates (GR) showed a seasonal cycle with a maximum in summer (Fig. 3). This is in agreement with previous analyses made for this site (Dal Maso et al., 2007; Yli-Juuti et al., 2011; Nieminen et al., 2014). In contrast to smaller sizes, in accumulation mode the median GRs had a minimum during summer.

The month-specific median growth rates were very similar in the nucleation and Aitken modes, varying between 1.8 and 4.1 nm/h. The highest growth rates, both in terms of the maximum values and on average, were observed in the accumulation mode. In wintertime, the growth rates in the accumulation mode were by a factor of 3 to 5 larger than in nucleation and Aitken modes, whereas in summer the median values were similar or slightly lower than at the smaller sizes.

## 3.2 Impacts of atmospheric conditions on growth rates

The coupling of the observed growth rates and the particle size is shown in Fig. 4, where the growth rate increases with an increasing mean diameter of the growing particle mode over the period of the observed growth. This was evident for all the





determined growth rates and for the long growth periods with duration more than 5 h (Fig. 4a), and for both winter and summer (Fig 4b). At diameters smaller than 30 nm, very few growth rates lower than 1 nm/h were observed. This is understandable, since with slow growth rates the coagulation scavenging decreases more effectively the concentrations of the nucleation mode particles (e.g. Kerminen and Kulmala, 2002), resulting in concentration levels at which our method may not

detect the growing mode anymore. We will inspect the impact of particle diameter on the growth rate later (Sect. 3.3). However, because in Fig. 4 the growth rate seems to be very different in different size ranges, in the following Section we inspect the impacts of other parameters on growth rate in 10 nm size bins.

**3.2.1 Impact of condensable vapour source on the growth rate**

The first source-related parameter that we inspected was the temperature. It has been shown that during the vegetation

growing season in Hyytiälä, the condensational growth of particles is driven by biogenic vapours, such as monoterpenes (Paasonen et al., 2013), and their emissions depend strongly on temperature (Günther et al, 1993). In Fig. 5 GR is depicted as a function of the mean temperature during the observed growth period in 10 nm size bins from below 10 nm to 200 nm in April-September. The growth rates clearly increased as a function of temperature in bins with diameters below 100 nm. In diameter bins of 100 – 130 nm the effect of temperature was not observed, but for bins with diameters > 130 nm a weak

negative correlation between GR and temperature was found.

We used linear least-squares fits in a log-linear space to examine the temperature dependence. Interestingly, the fitted functions, shown in each panel of Fig. 5 with fitting parameters and correlation coefficients tabulated in Appendix 1, were not very different for the diameter bins having the mean diameters lower than 100 nm. Instead of showing consistently higher growth rates for larger (or, closer to 100 nm) particles at certain temperature, Fig. 5 shows that growth periods

starting from larger sizes are observed on average with higher temperatures than those starting from smaller sizes. This could, in principle, suggest that the association between the particle diameter and growth rate depicted in Fig. 4 is not directly causal, but could stem from roughly same aged particles appearing at the measurement station at larger sizes in warmer air masses with higher concentrations of condensable vapours. We will examine this in more detail in Section 3.3.

Because of the relatively similar temperature dependences in size bins below 100 nm, all the growth rates in these bins

together show a reasonably clear connection with the temperature (Fig. 6a). The Pearson's correlation coefficients for $\log(GR)$ and temperature in April-September had $R = 0.20$ for the periods with the duration > 2 h and 0.35 for the periods with the duration > 5 h (the respective $p$-values, shown in Table 3, indicate that the correlations are statistically significant).

Next, we repeated the analysis by substituting the temperature with the monoterpene concentration proxy. Surprisingly, the correlation between GR and monoterpene concentrations was weaker (log-log correlation for April-September: $R = 0.18$

when duration > 2 h and 0.33 when duration > 5 h) than for GR and temperature. When the proxy for the monoterpene oxidation product concentrations $[MT_{Ox}]$ was applied, the correlations were even weaker (log-log correlations for the same as above: $R = 0.15$ and 0.26). However, a similar correlation test for GR with the oxidation rate of monoterpenes (OxRate)



revealed a stronger correlation (log-log correlations for the same as above: $R = 0.24$ and $0.39$) than for GR and temperature (Fig. 6b). The values of the linear least-square fits for growth rates as functions of temperature and monoterpene oxidation rate for growth periods starting in $d_P < 100$ nm are presented in Table 3. The linear least-square fitting parameters for GR as functions of monoterpene concentrations and oxidation rates were similar to those for GR as a function of temperature presented above (see Appendix 1).

Finally, we varied the weighting factors for OH and $NO_3$ oxidation reactions from 0.01 to 100. The highest correlation coefficients between GRs with the duration > 2 h and OxRate were obtained with weighting factors 3.8 for OH oxidation and 1.2 for $NO_3$ oxidation. Similar weighting factors for the duration > 5 h were 1.8 and 0.64, respectively. The resulting correlation coefficients were $R = 0.25$ for the duration > 2 h and $R = 0.40$ for the duration > 5 h. These are only 0.01 higher than the respective correlation coefficients for the oxidation rate without weighting factors, and thus the difference cannot be considered significant. Nevertheless, the fact that more diverse weighting factors could not be found, explains the observed lack of diurnal cycles in growth rates (Sect. 3.1). Since the major contributor to monoterpene oxidation rate during April–September is the ozonolysis reaction (see Figs. 9-10 in Kontkanen et al., 2016), which does not have a strong diurnal variation, the weighting factors with the observed magnitudes do not lead to an observable diurnal cycle in our long-term data. However, these results indicate that the contributions of hydroxyl and nitrate radicals, in addition to that of ozone, on the formation of condensable vapours should be included e.g. in modelling studies, and that it is important to include all the oxidizers in order not to overestimate the diurnal cycle of the growth rates.

### 3.2.2 Impact of condensation sink on the growth rates

A higher condensation sink is expected to decrease particle growth rates by consuming faster the condensable vapours. Thus, it is surprising that the observed particle growth rates correlated clearly better with the approximated oxidation rate of monoterpenes alone than with the same rate divided by CS, which would be the logical solution based on steady-state approximation of the condensable vapour concentration. However, there is a strong coupling between the temperature, monoterpene emissions and concentration of accumulation mode particles in many vegetated regions, including the forests around SMEAR II (Paasonen et al., 2013). This coupling stems from the enhanced growth of particles due to the higher temperatures and monoterpene emissions in the air mass history, which naturally leads to higher concentration of larger particles and thus higher CS (Liao et al., 2014). Due to this causality, the dependence between the observed growth rate and condensation sink, or rather its logarithm, is very similar to that between GR and temperature (Fig. 7): the negative relation between CS and GR is evident only in particle size ranges 110 nm $< d_P <$ 180 nm and in size ranges $d_P <$ 80 nm the correlation between GR and CS is positive.

The positive relation between GR and CS would indicate that the source of condensable vapours is closely connected to CS, which can result from the strong contribution of the (semi-)condensable vapours to the build-up of CS prior to the observation. Based on our data, this relation seems very strong. We were not able to find negative correlations between GR



at $d_P < 100$ nm and CS even for the subsets of data in which the diameter range and the range of monoterpene oxidation rate (representing our best estimate for the source of condensable vapours) were strictly constrained. A representative example can be found in Fig. 8, in the panel on the 3rd row from the top and the 4th column from the left. This seems intuitively difficult to understand. It is even more difficult to explain that the influence of GR on the build-up of CS overrules the

plausible decreasing impact of CS on GR in the Aitken mode, but not in the accumulation mode. Another possible explanation for our observation is that the condensation sink is not, for some reason, effective for the vapour(s) growing the nucleation and Aitken mode particles, indicating the importance of heterogeneous surface chemistry. Previously, Kulmala et al. (2017) discussed this kind of possibility when comparing the condensation sink and the required concentrations of vapours participating in new particle formation in a very different environment, Chinese mega-cities.

**3.2.3 Comparison of significance of influencing atmospheric parameters**

Since all the variables that were shown to correlate with the growth rate above are strongly interlinked, we tested which of them explains the variation of GR best in case the variation in the other parameters was limited. In Fig. 8 the relations of GR in the size range from 50 to 60 nm with the temperature, monoterpene concentration, monoterpene oxidation rate and condensation sink are presented by limiting the variation of one of these variables at a time to lie between its 30th and 70th

percentile. The highest correlation coefficients were found for GR as a function of monoterpene oxidation rate (3rd column from left) regardless of which of the other parameters was limited. Additionally, the lowest correlation coefficients in each column were encountered when the variation in the monoterpene oxidation range was limited (3rd row from up). Similar features were observed for different subsets of GRs in terms of the growth period duration, size range and time of the year, although not always as clearly as in the presented case. This finding confirms that the oxidation rate of monoterpenes is the

strongest of the inspected variables in determining particle growth rates.

It is to be notified that we also made an extensive number of tests with other variables recorded at the SMEAR II station (meteorological variables, gaseous and aerosol phase concentrations, ratios between different variables etc.) with similar methodologies as in Paasonen et al. (2010) and Kontkanen et al. (2016), but significant alternative or additional correlations were not found.

**3.3 Impact of particle diameter on growth rate**

The similarity of the functions fitted to GR vs temperature data in different size ranges below 100 nm (Fig. 5 and Appendix 1) could be interpreted so that the apparent relation between the diameter and GR (Fig. 4a) is caused by a link between temperature and the size in which the growth rate is observed. In order to investigate this further, we depict in Fig. 9 the growth rates as functions of the starting size of the observed growth period in different temperature ranges. The high end of

the GRs grows steadily with $d_P$ in all temperature ranges. The low end shows a similar increase when the GR starts at $d_P > 20$ nm. As discussed in Sect. 3.2 in relation to Fig 4a, the absence of data points at low GRs with $d_P < 20$ nm does not mean that


these growth rates do not exist, but that their observation may be impossible. This suggests that there is a direct connection between GR and particle size, which is inspected in more detail below.

We inspected the temporally overlapping growth periods, which are determined to take place simultaneously for at least one hour. In Fig. 10 the difference in the growth rates ($\Delta GR = GR(d_{P2}) - GR(d_{P2})$, where $d_{P2} > d_{P1}$) is depicted against the
difference in the mean diameter ($\Delta d_P = d_{P2} - d_{P1}$) at the starting moment of the overlap in growth periods. When CS was low or medium high for Hyytiälä (10a-c), the growth rate was on average higher for larger particles, and the correlation between $\Delta GR$ and $\Delta d_P$ was significant. This is in agreement with previous findings by Burkart et al. (2017), who analysed five days with simultaneous growth periods of different sized particles during Arctic marine observations.

However, when CS was higher than $4 \times 10^{-3}$ s$^{-1}$, the dependence seemed to disappear (Fig. 10d). This is another peculiarity
related to the condensation sink, which needs to be assessed in more detail in future studies, in addition to the opposite relation between GR and CS for particles in the Aitken and accumulation modes discussed in the previous Section. It should be noted that when the simultaneous growth periods were investigated in the temperature bins, the division to bins showing positive correlation between $\Delta GR$ and $\Delta T$ was not as clear as in Fig. 10.

### 3.3.1 Modelled particle growth rate due to semi-volatile partitioning

The diameter growth rate under a constant concentration of vapour should remain relatively constant with particle size at diameters larger than a few tens of nanometers (in which sizes the Kelvin effect does not affect the growth significantly) if the condensation is limited only by the condensation and evaporation of the vapour without any changes in the volatility of the vapour. The increase of GR with particle diameter suggests that the maximum uptake of semi-volatile vapours is influenced by aerosol-phase reactions, e.g. dimer formation, during which the volatility decreases. This has been earlier
proposed based on modelling e.g. by Apsokardu and Johnston (2018).

Our one particle process model, described in Sect 2.3 with atmospherically relevant input values for the base case (tabulated in Table 1), shows a clear increase in the diameter growth rate with an increasing particle diameter (blue solid line in Fig. 11) in roughly the same diameter range (10-300 nm) as the observations. This increase is caused by the aerosol-phase formation of non-volatile SVOC$_{dim}$, since the increase does not occur when the formation of these dimers was turned off (i.e. $k_{dim}$ set to
0, red line in Fig. 11a). The diameter at which the increase in GR starts, being between 10 and 20 nm, is determined by the Kelvin effect, since by setting Ke = 1 in Eq. (3) the increase appears immediately after 2 nm (yellow line in Fig 11a). When the diameter increases further, over 300 nm, GR starts to decrease. This is because, when the diameter increases and the particle approaches the continuum regime (Kn << 1, i.e. $d_P$ >> 150 nm), the Fuchs-Sutugin correction factor $\beta$ starts to decrease notably with an increasing diameter. This is demonstrated with the green line in Fig 11a, for which $\beta$ is set to
increase linearly with the diameter, similarly to the free-molecular regime (Kn >> 1, i.e. $d_P$ << 100 nm). In this case GR increases with an increasing diameter throughout the modelled sizes.



Figures 11b-d illustrate the sensitivity of the growth rate to gas phase concentrations of ELVOC and SVOC (Fig. 11b), SVOC saturation vapour concentration and dimerization rate coefficient (Fig. 11c), and the molar masses of ELVOC and SVOC (Fig. 11d). This sensitivity analysis gives us some suggestions for the parameters determining the particle growth rate in Hyytiälä:

- The diameter corresponding to maximum GR decreases with decreasing $C_{SVOC,sat}$, with increasing $k_{dim}$ and with decreasing $M_{SVOC}$. In our observations, we did not observe settling of the increase in GR when the diameter increased to over 200 nm. This suggests that the vapours mainly responsible for the particle growth in Aitken and accumulation mode would have either saturation vapour concentrations higher than $10^9$ cm$^{-3}$, $k_{dim}$ smaller than $1.66 \times 10^{-23}$ cm$^3$ s$^{-1}$ or molar masses higher than 300 g mol$^{-1}$.

- The growth rate at diameters below 10 nm is directly proportional to molar mass and concentration of ELVOC (assuming constant density). At larger diameters, the growth rate is directly proportional to SVOC concentration and inversely proportional to $C_{SVOC,sat}$, but it is less sensitive to SVOC molar mass. By comparing the GR values in Figs. 11b and 11d to Fig 3a, we estimate that the ELVOC concentration in Hyytiälä is typically below $1.6 \times 10^7$ cm$^{-3}$, assuming $M_{ELVOC} = 300$ g mol$^{-1}$. The highest SVOC concentrations seem to be around $3 \times 10^8$ cm$^{-3}$, assuming $C_{SVOC,sat} = 10^9$ cm$^{-3}$, and higher if the saturation vapour concentration is higher.

## 4 Conclusions

We generated an automatic method that seeks for growing particle modes from particle number size distribution data and determines the growth rate (GR) for these growth periods. This method finds growth periods from the nucleation mode ($d_P < 25$ nm) to the accumulation mode ($d_P > 100$ nm). We used the method to examine 20 years of particle size distribution data from a boreal forest observation site, SMEAR II, in Hyytiälä, Finland. All together 19513 growth periods of at least two hours of duration were determined, with the largest number of periods in the Aitken mode (10847).

The growth rates in the nucleation mode showed a clear annual cycle, with the highest rates being recorded in July and the lowest in December and January. A similar but less pronounced cycle was observed in the Aitken mode, but in the accumulation mode the annual cycle was opposite, having a minimum in July and August. Clear diurnal cycles were not observed.

We investigated the particle growth rates from April to September in more detail, since during this period the biogenic emissions are expected to dominate the aerosol growth. We found that the behaviour of the growth rates for particles smaller and larger than 100 nm were very different: in the nucleation and Aitken mode GR increased with an increasing temperature, while in the accumulation mode this relation was opposite. We showed that the temperature dependence of GR was likely caused by the formation of condensable vapours as GR correlated with the oxidation rate of monoterpenes stronger than with the temperature.



The growth rates were found to correlate in a similar way with the condensation sink (CS) as with the temperature and monoterpene oxidation rate, i.e. showing a positive correlation for GRs of particles with $d_P$ < 100 nm and negative correlations for the larger particles. On one hand, the positive correlations for the nucleation and Aitken mode particles are understandable, since the enhanced growth of particles leads to higher concentrations of accumulation mode particles, which

causes an increase in CS. On the other hand, it would be assumable that a higher CS would also have an opposite impact on the particle growth rate, since it should decrease the concentration of condensable vapours. This kind of an impact was not observed for particles with $d_P$ < 100 nm even when inspecting the relation between CS and GR under roughly constant monoterpene oxidation rates, which is our best estimate for the condensable vapour source. In the accumulation mode, GR decreased with an increasing CS in a similar manner to that of the temperature and monoterpene oxidation rate. One possible

interpretation of this is that the concentration of condensable vapours is not the limiting factor for the growth. Another possibility is that, for some reason, the vapours condensing on the nucleation and Aitken mode particles do not condense as efficiently onto larger particles. The latter interpretation is partly similar to the findings by Kulmala et al. (2017), who showed that in Chinese megacities the high condensation sink should prevent the observed new particle formation as nucleating vapours and small clusters should be effectively scavenged due to the very high values of CS.

Finally, we found that the maximum observed growth rate increased with an increasing particle diameter. While the highest observed growth rates at $d_P$ around 10 nm were roughly 10 nm/h, the highest growth rates increased steadily to around 30 nm/h for particles with $d_P$ of 100 nm, and this pattern continued in the accumulation mode. A similar result was found when comparing the growth rates of temporally overlapping growth periods, except for the cases where CS was high compared to the average CS at SMEAR II. We also showed with a single particle process model that the increase in GR as a function of

$d_P$ can be explained by the assumption that the growth is caused by the partitioning of semi-volatile vapours which, in the aerosol phase, form practically non-volatile dimers. This finding is in agreement with the modelling study by Apsokardu and Johnston (2018), as well as the observational study by Burkart et al. (2017) in the Arctic oceans. Our observations suggest that semi-volatile compounds might responsible for the particle growth to CCN sizes in continental environments as well.

Our study suggests that the aerosol growth to cloud condensation nuclei sizes in the boreal forest is dominated by the

condensational growth caused by semi-volatile oxidation products of biogenic volatile organic compounds. The observed increase in the particle growth rate as a function of particle size has a significant effect on the climate impacts of aerosol particles formed either during NPF events or emitted into the Aitken mode sizes from traffic or other sources. The increasing growth rate increases the fraction of the nucleation and Aitken mode particles surviving to CCN sizes and being able to form cloud droplets. This effect, or the processes leading to it, i.e. the semi-volatile vapours forming non-volatile dimers in the

aerosol phase, needs to be included in climate model simulations when aerosol-cloud and aerosol-radiation interactions are estimated. Additionally, the observation that the condensation sink appears not to limit the growth of particles in sub-CCN size range is in contrast with various estimates of the aerosol dynamics. Our findings suggest that the formation of CCN sized particles is not as strongly self-limiting process as previous studies have suggested.



**Appendix 1.**

Table A1. Fitting parameters resulting from linear least squares fits for parameterisations of growth rates with growth period starting sizes in 10 nm bins and the related correlation coefficient p-values, indicating the probability of getting similar correlation as random chance. Fittings and correlations are for all determined growth rates (> 2 h) during April-September.

5    Correlations which cannot be considered statistically significant ($p > 0.01$) are shaded.

| $d_P$ range (nm) | $GR = A + 10^{(B*T)}$ | | | $GR = 10^{(A+B*\log_{10}([MT]))}$ | | | $GR = 10^{(A+B*\log_{10}(OxRate))}$ | | | $GR = 10^{(A+B*\log_{10}(CS))}$ | | |
|---|---|---|---|---|---|---|---|---|---|---|---|---|
| | $B$ | $A$ | $p$-value | $B$ | $A$ | $p$-value | $B$ | $A$ | $p$-value | $B$ | $A$ | $p$-value |
| <10 | 0,015 | -3,9 | $1*10^{-08}$ | 0,24 | -2,0 | $1*10^{-07}$ | 0,28 | -1,2 | $2*10^{-10}$ | 0,17 | 0,8 | $3*10^{-03}$ |
| 10-20 | 0,011 | -2,7 | $2*10^{-08}$ | 0,30 | -2,5 | $2*10^{-14}$ | 0,28 | -1,1 | $5*10^{-14}$ | 0,19 | 0,9 | $1*10^{-05}$ |
| 20-30 | 0,007 | -1,6 | $3*10^{-05}$ | 0,18 | -1,3 | $1*10^{-07}$ | 0,18 | -0,6 | $2*10^{-07}$ | 0,30 | 1,2 | $5*10^{-11}$ |
| 30-40 | 0,015 | -3,8 | $1*10^{-12}$ | 0,26 | -2,1 | $2*10^{-13}$ | 0,32 | -1,4 | $9*10^{-18}$ | 0,41 | 1,5 | $2*10^{-18}$ |
| 40-50 | 0,027 | -7,3 | $1*10^{-27}$ | 0,40 | -3,5 | $1*10^{-21}$ | 0,49 | -2,4 | $3*10^{-30}$ | 0,60 | 1,9 | $4*10^{-24}$ |
| 50-60 | 0,030 | -8,2 | $2*10^{-22}$ | 0,41 | -3,6 | $3*10^{-14}$ | 0,49 | -2,5 | $5*10^{-19}$ | 0,54 | 1,7 | $4*10^{-14}$ |
| 60-70 | 0,029 | -7,8 | $8*10^{-17}$ | 0,34 | -2,9 | $4*10^{-07}$ | 0,65 | -3,4 | $5*10^{-20}$ | 0,68 | 2,0 | $2*10^{-16}$ |
| 70-80 | 0,024 | -6,3 | $3*10^{-08}$ | 0,17 | -1,3 | $3*10^{-02}$ | 0,47 | -2,3 | $1*10^{-07}$ | 0,28 | 1,1 | $5*10^{-03}$ |
| 80-90 | 0,018 | -4,8 | $3*10^{-05}$ | 0,08 | -0,4 | $3*10^{-01}$ | 0,25 | -1,0 | $1*10^{-02}$ | 0,22 | 0,9 | $9*10^{-02}$ |
| 90-100 | 0,016 | -4,2 | $5*10^{-04}$ | -0,07 | 1,1 | $4*10^{-01}$ | 0,01 | 0,3 | $9*10^{-01}$ | -0,06 | 0,3 | $6*10^{-01}$ |
| 100-110 | -0,001 | 0,7 | $8*10^{-01}$ | -0,22 | 2,6 | $7*10^{-03}$ | -0,17 | 1,4 | $9*10^{-02}$ | -0,20 | 0,0 | $4*10^{-02}$ |
| 110-120 | 0,004 | -0,8 | $4*10^{-01}$ | -0,12 | 1,6 | $2*10^{-01}$ | -0,01 | 0,5 | $1*10^{00}$ | -0,16 | 0,1 | $1*10^{-01}$ |
| 120-130 | 0,008 | -1,9 | $1*10^{-01}$ | 0,00 | 0,5 | $1*10^{00}$ | -0,03 | 0,6 | $8*10^{-01}$ | -0,27 | -0,2 | $5*10^{-03}$ |
| 130-140 | -0,008 | 2,7 | $1*10^{-01}$ | -0,24 | 2,8 | $1*10^{-02}$ | -0,20 | 1,6 | $2*10^{-02}$ | -0,28 | -0,2 | $7*10^{-03}$ |
| 140-150 | -0,026 | 7,9 | $3*10^{-07}$ | -0,45 | 4,8 | $6*10^{-07}$ | -0,45 | 3,0 | $5*10^{-08}$ | -0,54 | -0,9 | $3*10^{-09}$ |
| 150-160 | -0,019 | 5,8 | $1*10^{-04}$ | -0,40 | 4,4 | $7*10^{-05}$ | -0,42 | 2,9 | $2*10^{-06}$ | -0,44 | -0,6 | $1*10^{-05}$ |
| 160-170 | -0,013 | 4,3 | $1*10^{-03}$ | -0,26 | 3,1 | $9*10^{-04}$ | -0,33 | 2,4 | $4*10^{-06}$ | -0,31 | -0,2 | $8*10^{-05}$ |
| 170-180 | -0,012 | 4,1 | $3*10^{-02}$ | -0,23 | 2,8 | $2*10^{-02}$ | -0,27 | 2,1 | $5*10^{-03}$ | -0,39 | -0,4 | $5*10^{-04}$ |
| 180-190 | -0,016 | 5,1 | $3*10^{-02}$ | -0,27 | 3,1 | $5*10^{-02}$ | -0,41 | 2,8 | $9*10^{-04}$ | -0,28 | -0,2 | $4*10^{-02}$ |
| 190-200 | -0,001 | 0,7 | $9*10^{-01}$ | 0,09 | -0,3 | $4*10^{-01}$ | -0,17 | 1,5 | $2*10^{-01}$ | -0,31 | -0,2 | $4*10^{-02}$ |

**Acknowledgements**

This study was funded by the Academy of Finland (project no. 307331), European Commission (project ID: 742206), the
10   Doctoral Programme in Atmospheric Sciences (ATM-DP, University of Helsinki, Jenni Kontkanen), and the European




Regional Development Fund and the Mobilitas Pluss programme (project MOBTT42). The authors would like to thank Dr. Santtu Mikkonen and Mr. Ville Leinonen from the University of Eastern Finland for fruitful discussion.

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



Table 1. Values of the variables in the model runs for the base case run. Variables marked with asterisk ($^*$) are varied in the sensitivity analysis. The variables mainly related to the atmospheric conditions and vapours are shown on the left-hand side and those related to the growing particle on the right-hand side.

| | | | |
|---|---|---|---|
| $C_{ELVOC}$ (# cm$^{-3}$)$^*$ | $4\times10^6$ | Initial [ELVOC] (#) | 6 |
| $C_{SVOC}$ (# cm$^{-3}$)$^*$ | $4\times10^7$ | Initial [SVOC] (#) | 0 |
| $C_{SVOC,sat}$ (# cm$^{-3}$)$^*$ | $1\times10^9$ | Initial [SVOC$_{dim}$] (#) | 0 |
| $V_{mol}$ (m$^3$ mol$^{-1}$) | $1.4\times10^{-4}$ | Initial $d_P$ (nm) | 2 |
| $M_{ELVOC}=M_{SVOC}$ (g mol$^{-1}$)$^*$ | 300 | $\sigma$ (N m$^{-1}$) | 0.08 |
| $M_{SVOC_{dim}}$ (g mol$^{-1}$)$^*$ | 600 | $K_{dim}$ (cm$^3$ s$^{-1}$)$^*$ | $1.66\times10^{-23}$ |
| $T$ (K) | 288 | $\rho$ (g cm$^{-3}$) | 1.5 |
| $D$ (cm$^2$ s$^{-1}$) | 0.1 | | |

Table 2. Number of determined growth periods segregated by time of year (rows), times of day (columns) and aerosol size modes (top-left: nucleation mode, middle: Aitken mode, bottom-right: accumulation mode). Note that the segregation to modes is made based on the starting size of the observed growth period.

| | 0-6 hrs | 6-12 hrs | 12-18 hrs | 18-24 hrs | **SUM** |
|---|---|---|---|---|---|
| Mar-May | 212 | 280 | 576 | 356 | **1424** |
| | 1109 | 681 | 569 | 503 | **2862** |
| | 399 | 308 | 331 | 218 | **1256** |
| Jun-Aug | 16 | 142 | 181 | 108 | **447** |
| | 1129 | 802 | 646 | 673 | **3250** |
| | 403 | 296 | 361 | 281 | **1341** |
| Sep-Nov | 86 | 105 | 285 | 193 | **669** |
| | 960 | 548 | 663 | 481 | **2652** |
| | 427 | 306 | 292 | 238 | **1263** |
| Dec-Feb | 112 | 98 | 180 | 103 | **493** |
| | 745 | 425 | 503 | 410 | **2083** |
| | 597 | 445 | 443 | 288 | **1773** |
| **SUM** | **426** | **625** | **1222** | **760** | 3033 |
| | **3943** | **2456** | **2381** | **2067** | 10847 |
| | **1826** | **1355** | **1427** | **1025** | 5633 |




Table 3. Parameters of linear least square fits for growth rates starting from $d_P < 100$ nm as a function of temperature (first row) and monoterpene oxidation rate (second row), and the related correlation coefficients and $p$-values. Upper values are for growth periods with the duration $> 2$ h and the lower values in Italics for the duration $> 5$ h.

| | April-September | | | | Whole year | | | |
| --- | --- | --- | --- | --- | --- | --- | --- | --- |
| | $B$ | $A$ | $R$ | $p$ | $B$ | $A$ | $R$ | $p$ |
| $GR = A + 10^{B \times T}$ | 0.014 | -3.7 | 0.20 | $10^{-66}$ | 0.0067 | -1.6 | 0.13 | $10^{-51}$ |
| | *0.017* | *-4.3* | *0.35* | *$10^{-44}$* | *0.0099* | *-2.4* | *0.29* | *$10^{-45}$* |
| $GR = 10^{A + B \times log_{10} \times OxRate}$ | 0.30 | -1.3 | 0.24 | $10^{-83}$ | 0.18 | -0.69 | 0.18 | $10^{-83}$ |
| | *0.34* | *-1.5* | *0.39* | *$10^{-47}$* | *0.24* | *-0.96* | *0.34* | *$10^{-53}$* |



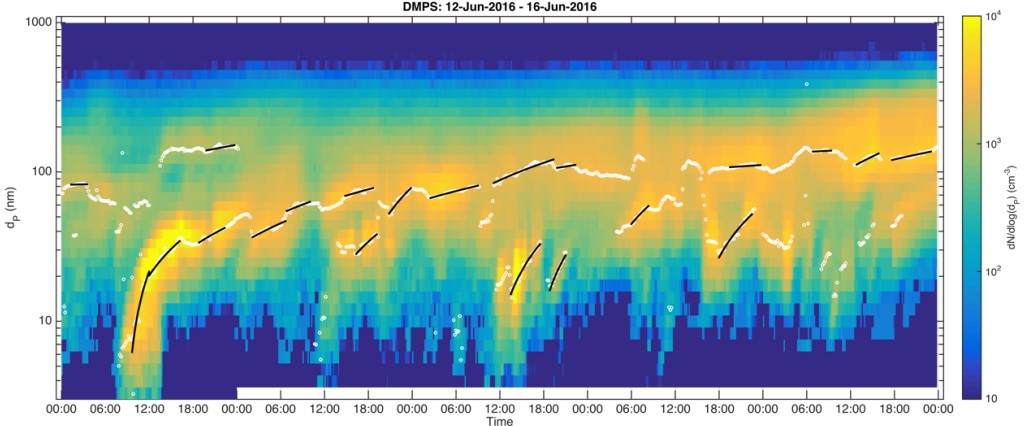

**Figure 1. An example of the evolution of particle size distribution and the determined growth periods over five consecutive days in June 2016. White circles show the found peaks in particle size distribution and the black lines show the determined monotonic growth periods.**

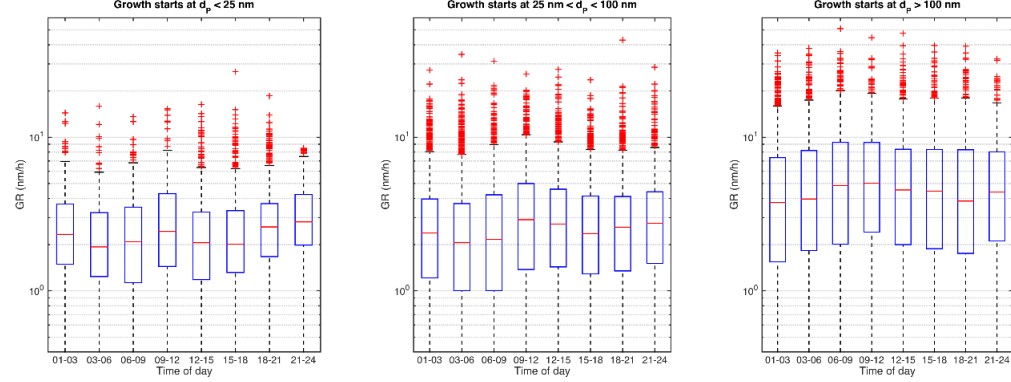

**Figure 2. Diurnal variation of all the determined growth rates in different size ranges. Red horizontal line represents the median value and the blue box the 25$^{th}$ and 75$^{th}$ percentile values. The whiskers reach approximately +/- 2.7σ and the red markers are outliers from this range.**





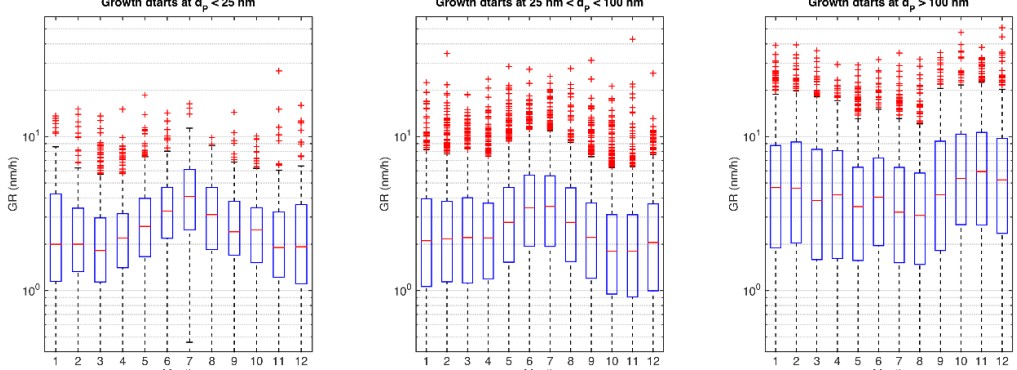

**Figure 3. Monthly variation of all the determined growth rates in different size ranges. See caption for Fig. 2 for details of the markers.**

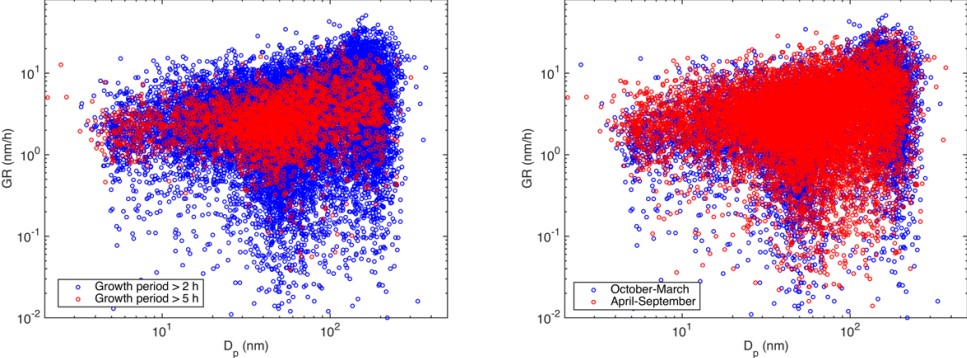

5     **Figure 4. Observed particle growth rate as a function of the initial size of the growing mode, in panel a) separated with the length of the observed growth period and in panel b) with the time of the year.**





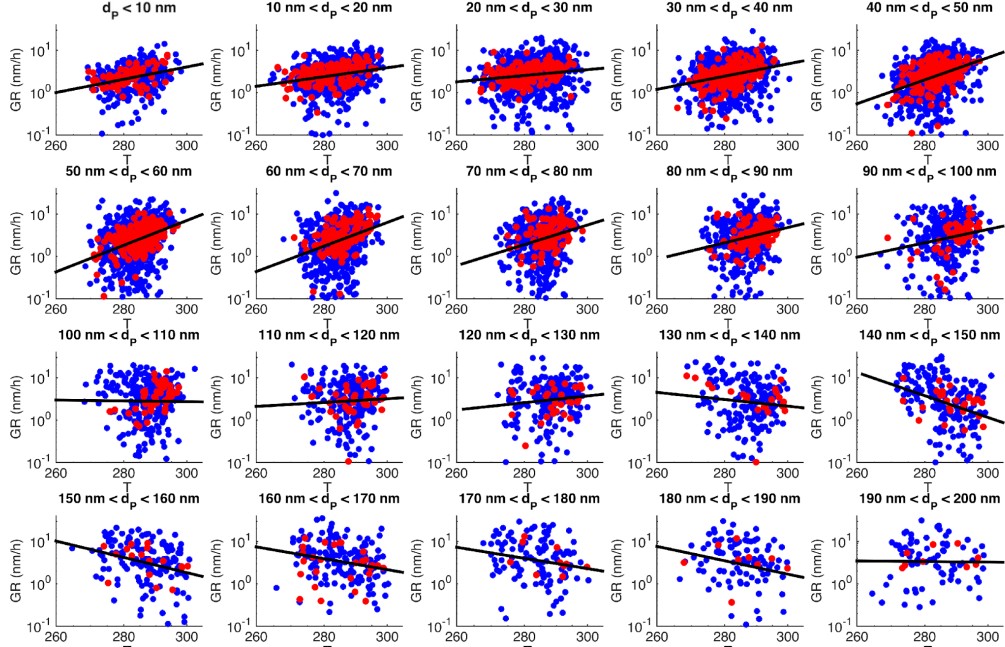

**Figure 5. Particle growth rate as a function of mean temperature during the growth period, binned with respect to the start size of the observed growth. The blue points depict all the determined growth periods, red ones the long (> 5 h) growth periods and the black lines are log-linear least squares fittings for all the growth periods (blue points).**





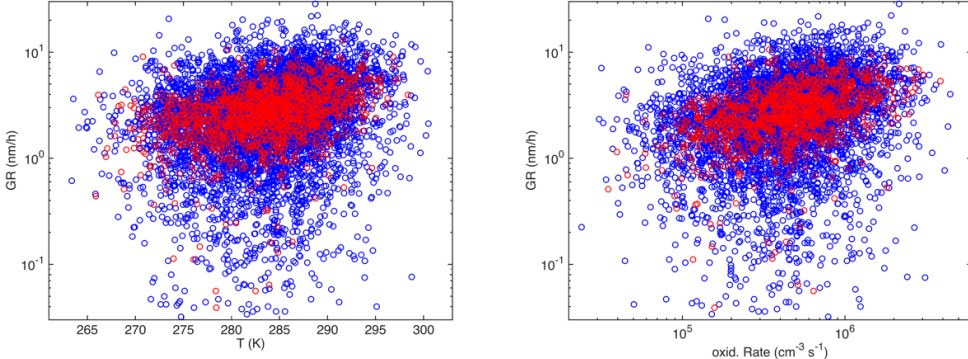

**Figure 6.** Particle growth rate (April-September, growth starts at $d_P$ <100 nm) as a function of temperature (a) and oxidation rate of monoterpenes (b). Blue circles are for growth period duration > 2 h and red for duration > 5 h.

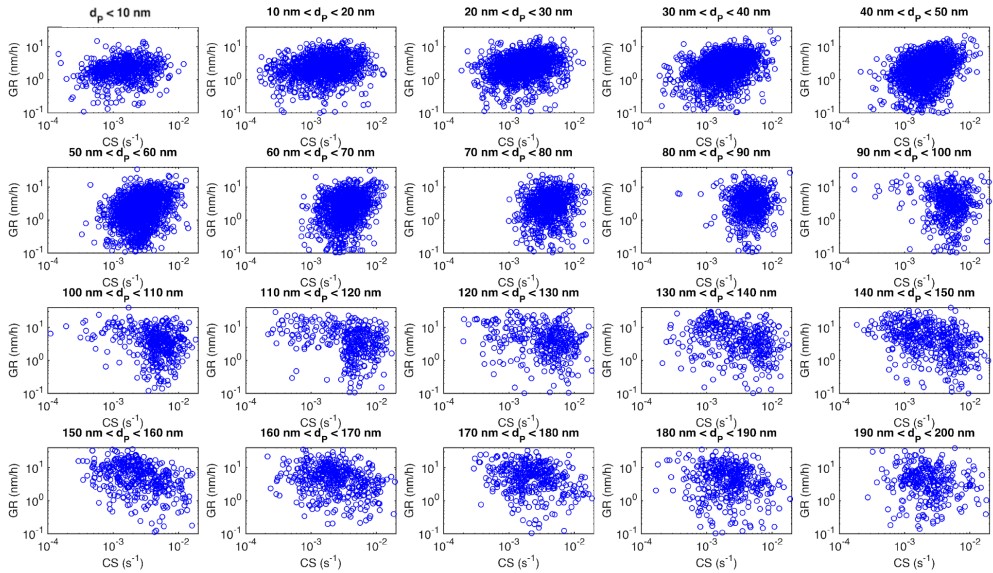

**Figure 7.** Growth rates with duration > 2 h during April-September as a function of condensation sink in size bins.





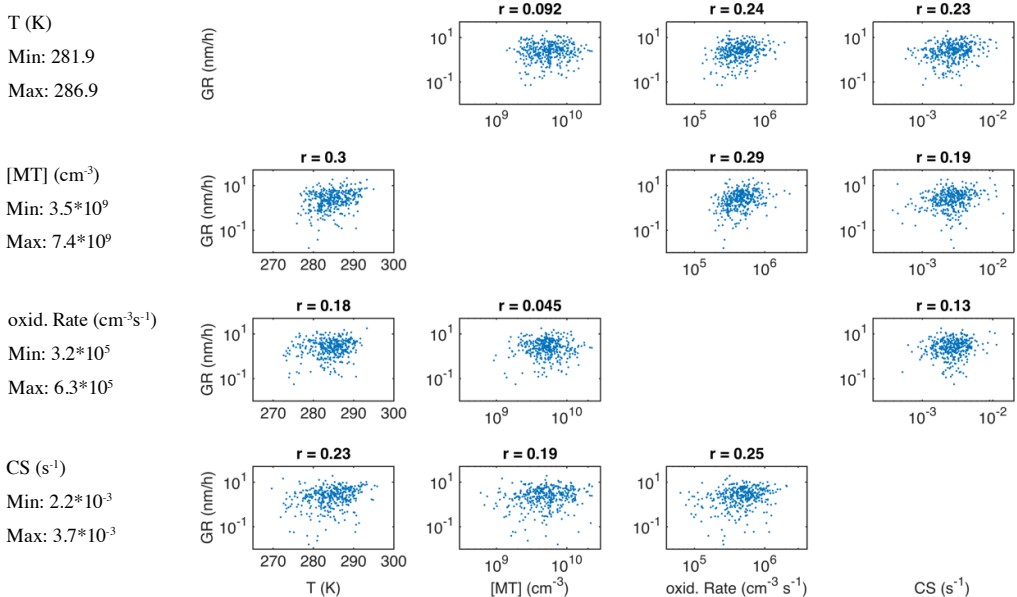

**Figure 8.** Growth rate of growth periods with duration > 2 h and starting size 50 nm < $d_P$ < 60 nm during April-September depicted as a function of temperature (1st column), monoterpene concentration (2$^{nd}$ column), monoterpene oxidation rate (3$^{rd}$ column) and condensation sink (4$^{th}$ column), while one of these four variables is limited to vary between its 30$^{th}$ and 70$^{th}$ percentile (limited variable for each row and the percentile values indicated on the left-hand side).



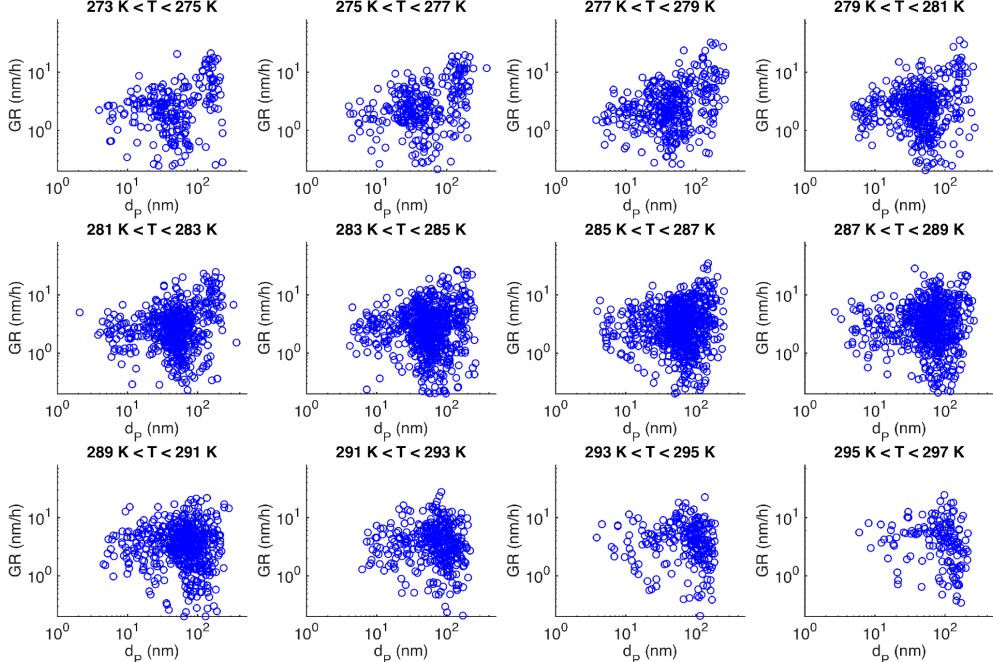

**Figure 9. Growth rate as a function of particle diameter for growth periods with duration > 2 h in April-September, presented in temperature bins.**





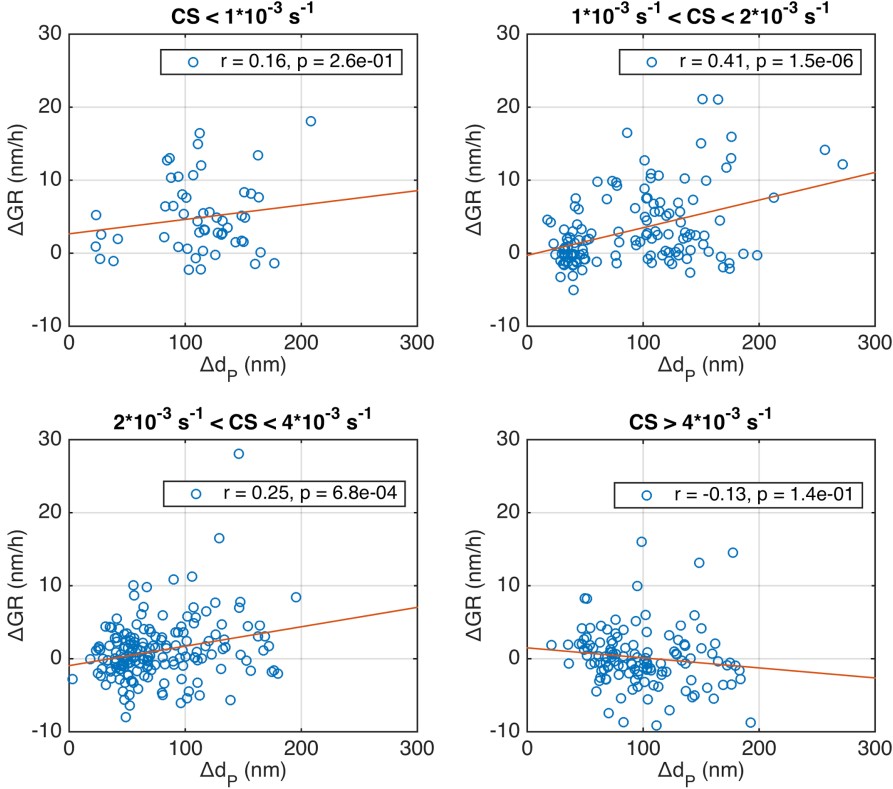

**Figure 10. Difference in growth rate as a function of difference in diameter for growth periods that overlap temporally for at least an hour. Data are presented in different condensation sink ranges for April-September.**





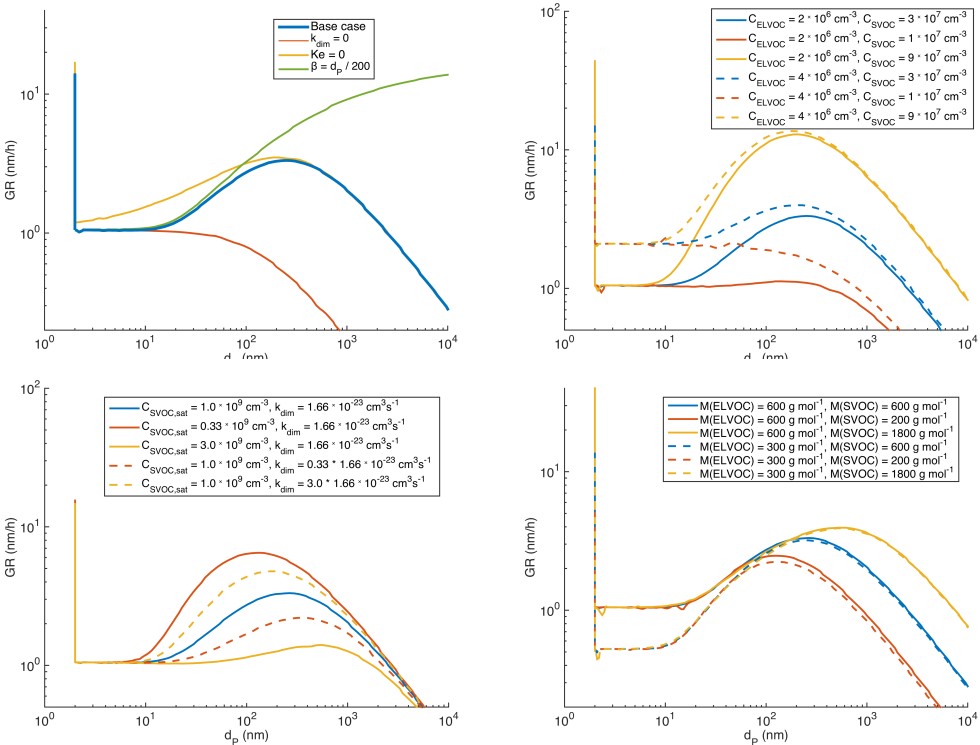

**Figure 11. Modelled growth rate of an aerosol particle with indication of factors causing the changes in GR as a function of diameter (panel a) and sensitivity analysis towards indicated factors (panels b-d). More details in text.**

