# Peer review of "Comprehensive analysis of particle growth rates from nucleation mode to cloud condensation nuclei in Boreal forest"

_Atmospheric Chemistry and Physics, 2018_

## Referee Comment (RC1) · Anonymous Referee #2 · 4 Apr 2018

This is a nice manuscript that analyzes particle size distribution (PSD) data acquired over a 20 year period in a Boreal forest. The procedure allows growth rates (GR) to be determined over a broad range of particle diameters spanning nucleation, Aitken and accumulation modes. The most significant finding is that Aitken mode particles grow at a faster rate than nucleation mode particles, a situation that is not fully captured by regional or global models, but has significant implications for production of CCN. This study adds to a growing body of work suggesting that chemical reactions within the particle volume can enhance GR.

The manuscript is well written and easy to follow. There are a few topics that the

authors could discuss a bit further.

Base case parameters for the one-particle process model are given in Table 1, and simulations using a range of values around the base values are shown in Figure 11. I assume that the base values and ranges for ELVOC and SVOC were chosen to be consistent with APi-TOF data from ambient and/or laboratory measurements. How was the base value for Kdim (dimerization rate constant) selected? Was it simply chosen numerically to give calculated GR of the same order of magnitude as those extracted from the experimental PSD data?

Bottom of page 7. Would it be more accurate in this sentence to say that for Figure 4, the maximum GR (or the average GR for data points > 1 nm/hr) increases with increasing diameter? This particle size dependence is not observed for GR < 1 nm/hr owing to the fact that low GR are hard to detect for small particles, since in these instances the particles are more likely to be lost by coagulation (page 8 lines 2-4). Because smaller GR are more likely to be observable as the particle diameter increases, could this effect be the source of the weaker correlations observed for accumulation vs. nucleation/Aitken particles in Figures 5, 7 and especially 10?

Alternatively, could the weaker correlations of accumulation particles simply be a consequence of the uncertainty associated with GR measurement as a function of beginning particle diameter? For example, it would seem to be much easier (more accurate and precise) to measure a 1 nm/hr GR for particles beginning at 50 nm than 200 nm since the relative change in the diameter is so much greater for smaller particles.

---

## Referee Comment (RC2) · Anonymous Referee #1 · 5 Apr 2018

The authors present a new automated method to derive particle growth rates from size-distribution data even in situations where no direct new particle formation is observed. They apply the method to an impressive 20 year DMPS-dataset taken at Hyytiälä, Finland. With this approach they achieve to get insights into particle growth for nucleation, Aitken and accumulation mode particles. They clearly show that the oxidation rate of monoterpenes is an important parameter for growth in a boreal forest site and their findings support theories of the importance of reactive uptake, especially for Aitken mode particles, where they find generally higher growth rates as in the nucleation mode.

I congratulate the authors for the well-designed automated growth rate method and the

impressive analysis of a huge dataset including some very interesting findings. The manuscript is comprehensive and well-written, but needs some technical clean-up to make it even more reader-friendly. Moreover, I have some minor comments, which the authors should address before I can recommend publication in ACP.

**Questions/Request for clarification:**

- Page 5, line 5 I think it would be helpful for the reader if you quantify the typical number of n, i.e. PSD measurements per day, or at least the time-resolution of the DMPS system. This would help the reader to identify how many PSDs usually fall in the range for a GR determination, or how strong the smoothing by the five-time-step-median filter actually is.

- In my opinion, the current manuscript does not really discuss, how well the new method actually works and what limitations it has. Is it for example catching most growth periods which were analyzed classically as they follow NPF? Additionally, I think the authors should clarify that the method only infers apparent growth rates, which might cause problems if it is applied to heavier polluted environments. For example coagulation within the growing population might mimic condensational growth and this is not captured by this method. Kuang et al. 2012 (ACP) and Pichelstorfer et al. 2018 (ACP) developed methods which take such effects into account, however they did not yet demonstrate to work with this kind of DMPS data sets.

- Section 2.3 and especially Table 1. I very much appreciate the simplicity of the model, but it seems to me that it was tuned a bit to fit the results. In Table 1, the molecular volume V does not correspond to M/rho, why? The surface tension of 0.08 is by more than a factor of 2 higher than values usually assumed for organics (see e.g. Tröstl et al. 2016, (Nature) ) and bigger than to one of water. This leads to a significantly increased Kelvin-diameter of roughly 12 nm. As a consequence

the range when the effects of SVOC dimerization start to be important is set to larger diameters. It would be good if you could specifiy why the values were chosen that way. Also, e.g. Kdim lacks any explanation.

- Fig. 4 and Section 3.2. Whenever the authors correlate the GR with a particle size they use the initial size of the growth. Growth rates are inferred from a minimum size to a maximum size, and as GR and the observed growth period varies as the authors point out in Sec. 2.2 I would assume that the mean size of the growth rate measurement gives a more representative value for the diameter where the GR is actually observed.

**Technical corrections:**

- Please consider to cleanup your Figures. Generally I recommend using bigger axis ticks to make the axis better readable. Additionally, while, e.g. Figure 10 has very well readable axis labels, this is not the case for Figures 1-4 and 6.

- Please check carefully the usage of definite articles, e.g. p.2 l.8 "by condensation growth", p.2 l.11 "the importance of growth", p2. l.13 "fraction of CCN originating from growth of smaller particles", p.8 l.15 "that we inspected was temperature", p.10 l.27 "50 to 60 nm with temperature, monoterpene concentration", etc.

- Page 3, lines 6-8. I would point towards Tröstl et al., 2016 (Nature), because they directly describe the Kelvin effect for organics and its influence on growth.

- Fig.5, Fig.7 and Fig. 9 I am just wondering, if a reduction of used bins would make the Figures far easier to read and understand, without losing the main conclusions.

- Page 9, l. 6-15. This paragraphs lacks a conclusion. Monoterpene concentrations are expected to have a weaker correlation than temperature, as temperature

not only controls the emissions but also the reaction rates. Given the negative correlation found with the CS and discussed in Sec. 3.2.2. it seems to be logical that the correlation with monoterpene oxidation rate is the strongest. This could be pointed out.

- Page 12, l. 13 and Fig. 11 a. While in the text Ke is set to 1 the Figure legend says Ke=0.

---

## Author Response (AR1)

*This is a nice manuscript that analyzes particle size distribution (PSD) data acquired over a 20 year period in a Boreal forest. The procedure allows growth rates (GR) to be determined over a broad range of particle diameters spanning nucleation, Aitken and accumulation modes. The most significant finding is that Aitken mode particles grow at a faster rate than nucleation mode particles, a situation that is not fully captured by regional or global models, but has significant implications for production of CCN. This study adds to a growing body of work suggesting that chemical reactions within the particle volume can enhance GR. The manuscript is well written and easy to follow. There are a few topics that the authors could discuss a bit further.*

We thank the Referee #2 for the positive feedback and good suggestions for sharpening the manuscript further.

*Comment 1.*
*Base case parameters for the one-particle process model are given in Table 1, and simulations using a range of values around the base values are shown in Figure 11. I assume that the base values and ranges for ELVOC and SVOC were chosen to be consistent with APi-TOF data from ambient and/or laboratory measurements. How was the base value for Kdim (dimerization rate constant) selected? Was it simply chosen numerically to give calculated GR of the same order of magnitude as those extracted from the experimental PSD data?*

This is a good point by the referee. The value for Kdim is taken from Apsokardu and Johnston (2018), who based their value on a study by Ervens and Volkamer (2010). These references were unintentionally not given in the manuscript. Both are now added to the revised manuscript.

*Comment 2.1*
*Bottom of page 7. Would it be more accurate in this sentence to say that for Figure 4, the maximum GR (or the average GR for data points > 1 nm/hr) increases with increasing diameter? This particle size dependence is not observed for GR < 1 nm/hr owing to the fact that low GR are hard to detect for small particles, since in these instances the particles are more likely to be lost by coagulation (page 8 lines 2-4).*

The referee is correct here. We actually mentioned especially the increase of the maximum GR as a function of increasing diameter when discussing the Fig. 9, but it is true that it is better to discuss this more exactly already related to Fig. 4. We modified the first sentences of this section to be as follows:

"The coupling of the observed growth rates and the particle size is shown in Fig. 4. Especially the highest observed growth rates increase when the mean diameter of the growing particle mode increases, but a similar increase is observed also for the lowest growth rate values for diameters larger than 30 nm. These features are evident for all the determined growth rates and for the long growth periods with duration more than 5 h (Fig. 4a), and for both winter and summer (Fig 4b)."

*Comment 2.2*
*Because smaller GR are more likely to be observable as the particle diameter increases, could this effect be the source of the weaker correlations observed for accumulation vs. nucleation/Aitken particles in Figures 5, 7 and especially 10?*

We do not think this should be the reason for weaker correlations in the accumulation mode, at least not the only one. Actually, one could also expect the opposite, because the correlations should be easier to determine when the growth rate varies more i.e. when also the slow growth rates can be detected. Furthermore, even though the correlations in the accumulation mode are weaker, they are, especially in size ranges from 140 to 170 nm, statistically significant (see Appendix 1 in the manuscript). And finally, the analysis shows negative correlations in accumulation mode, instead of positive as in smaller size ranges. Based on these reasons, we find that the lack of observed low growth rates in nucleation mode is not the reason for weaker correlations in the accumulation mode, in comparison to those in smaller modes.

*Comment 2.3*
*Alternatively, could the weaker correlations of accumulation particles simply be a consequence of the uncertainty associated with GR measurement as a function of beginning particle diameter? For example, it would seem to be much easier (more accurate and precise) to measure a 1 nm/hr GR for particles beginning at 50 nm than 200 nm since the relative change in the diameter is so much greater for smaller particles.*

This is a very good remark by the referee. Even in terms of the higher end of the GR values, the GRs increase by only a factor of 3 while the diameter increases by a factor of 10. Since the DMPS size bins have more or less similar relative width, the bin width also increases by a factor of 10 in this diameter change. Thus, it is very probable that the uncertainties, also relative ones, in GRs at larger diameters are larger than those in smaller diameters. We added the following sentences to the end of the first paragraph of Section 3.2.1:

"It should be noted that the uncertainties in the determined values of growth rates increase with an increasing diameter, because the relative change in diameter is larger for smaller particles. Another factor contributing to higher uncertainties for larger GRs is that the width of the DMPS size channels is roughly directly proportional to the diameter. Thus, the growth rates at larger diameters are determined with coarser particle size distributions relative to the growth rates, which increase at most by a factor of 3 when the diameter increases by a factor of 10 (in Fig. 4, the higher end of GRs increases from ~7 nm/h at 10 nm to 20 nm/h at 100 nm)."

**Anonymous Referee #1**

*The authors present a new automated method to derive particle growth rates from size-distribution data even in situations where no direct new particle formation is observed. They apply the method to an impressive 20 year DMPS-dataset taken at Hyytiälä, Finland. With this approach they achieve to get insights into particle growth for nucleation, Aitken and accumulation mode particles. They clearly show that the oxidation rate of monoterpenes is an important parameter for growth in a boreal forest site and their findings support theories of the importance of reactive uptake, especially for Aitken mode particles, where they find generally higher growth rates as in the nucleation mode. I congratulate the authors for the well-designed automated growth rate method and the impressive analysis of a huge dataset including some very interesting findings. The*
*manuscript is comprehensive and well-written, but needs some technical clean-up to make it even more reader-friendly. Moreover, I have some minor comments, which the authors should address before I can recommend publication in ACP.*

We thank the referee for very valuable comments and suggestions.

*Questions/Request for clarification:*
*• Page 5, line 5 I think it would be helpful for the reader if you quantify the typical number of n, i.e. PSD measurements per day, or at least the time-resolution of the DMPS system. This would help the reader to identify how many PSDs usually fall in the range for a GR determination, or how strong the smoothing by*
*the five-time-step-median filter actually is.*

This is correct, in the revised manuscript we express the time resolution of the DMPS measurements (10 min) in Sect. 2.1 (page 3, line 25) and add "(i.e. $PSD_n$ with $1 < n < 7$)" in the sentence denoted by the reviewer:
"If the first peak is determined in $PSD_0$, the timewise closest peak in $PSD_{n>0}$ is added to the same group, if it takes place within an hour from $PSD_0$ (i.e. $PSD_n$ with $1 < n < 7$) and is close enough in

size (maximum allowed difference is 10 nm for peaks with 5 $d_P$ < 50 nm and 50 nm for peaks with $d_P$ > 150 nm).")

• *In my opinion, the current manuscript does not really discuss, how well the new method actually works and what limitations it has. Is it for example catching most growth periods which were analyzed classically as they follow NPF?*

This is true. We decided to make a small comparison to manually analyzed growth rates after NPF events. We received GRs and their start and end times in size range 3-25 nm in Hyytiälä during 2003-2013 (Nieminen et al., 2014) from Dr. Tuomo Nieminen and compared our results with those. In the data we received there were 153 manually determined GRs, for which the start and end times were available. Out of these 153 manually determined GRs, our method captured 73 % (111 growth periods). The GRs determined with the automatic method also correlated well with the manual GRs (R = 0,81). The comparison of the growth rates is presented in Fig. R1 below. We find this accuracy to be reasonably good, since our method was not developed for determining growth rates especially in the nucleation mode, but in Aitken and accumulation modes. In the manual determination, the selection of peaks in particle size distribution data (white circles in Fig. 1) from which the GR is determined, is made visually and human eye can naturally connect more information for verifying the reliability of the determined GR than our automatic method. It should be also noticed that, since the manual method relies on visual inspection of the data, exactly similar results would not be expected from different persons using the exactly similar manual method.

We added the following sentences to the Methods Sect. 2.2 of the manuscript:
"We made a comparison between GRs determined with our automatic method and manually determined GRs for nucleation mode particles (Nieminen et al., 2014). For the comparison, we received start and end times of 153 growth periods during years 2003-2013. It is notable that the manual growth rates were determined only for the time until the mode reaches 25 nm in diameter, because the initial purpose for their determination had been calculating the new particle formation rates, whereas the compared automatic GRs were for growth periods, which had initial diameters below 25 nm. In order to prevent the possibility of comparing different growth parts of a growth period during which the growth rate would have drastically changed, we chose for comparison only the growth periods for which the automatic and manual growth periods overlapped for at least two hours. Another note to be made on the manual GR data is that these 153 events represent only a small fraction of the manual GR values for the years 2003-2013, but for the rest of the manual GRs only the dates (without start and end times) were readily available."

To the Results (Sect. 3.1) we added the following sentences:
"The comparison of nucleation mode GRs with manually determined GRs from Nieminen et al. (2014) showed a strong correlation (R = 0.81) between automatic and manual GRs. Out of the 153 manually determined growth periods our method found 111, equaling to 73 %. In 93 % of the growth periods detected with both methods, the automatic GR was within a factor of two, and in 76 % within a factor of 1.5 from the manually determined GR. We find this accuracy to be reasonably good, since our method was not developed for determining growth rates especially in the nucleation mode, but in Aitken and accumulation modes. In the manual determination, the selection of peaks in particle size distributions (white circles in Fig. 1) from which the GR is

determined, is made visually and human eye can naturally connect more information for verifying the reliability of the determined GR than our automatic method. It should be also noticed that, since the manual method relies on visual inspection of the data, exactly similar results would not be expected from different persons using the exactly similar manual method."

[Figure]

Figure R1, not added to the manuscript. Comparison of automatically determined growth rates and manual growth rates determined for growth periods that overlapped at least for 2 hours. More details in text above.

• *Additionally, I think the authors should clarify that the method only infers apparent growth rates, which might cause problems if it is applied to heavier polluted environments. For example coagulation within the growing population might mimic condensational growth and this is not captured by this method. Kuang et al. 2012 (ACP) and Pichelstorfer et al. 2018 (ACP) developed methods which take such effects into account, however they did not yet demonstrate to work with this kind of DMPS data sets.*

This is also true. We added to the Sect. 2.2 the following:
"It should be noted that our method simply searches for monotonic increases of particle mode diameters, it does not differentiate the condensational growth from growth due to coagulation or possible other phenomena that may cause apparent growth of a particle mode. Such phenomena, e.g. the faster coagulation scavenging of the smallest particles within a mode in comparison to the largest particles within the same mode, are typically considered more significant for particle growth in diameter ranges below 10 nm and in more polluted environments. Thus, we assume that the results in this article are not significantly impacted by them. "

• *Section 2.3 and especially Table 1. I very much appreciate the simplicity of the model, but it seems to me that it was tuned a bit to fit the results. In Table 1, the molecular volume V does not correspond to M/rho, why? The surface tension of 0.08 is by more than a factor of 2 higher than values usually assumed for organics (see e.g. Tröstl et al. 2016, (Nature) ) and bigger than to one of water. This leads to a significantly increased Kelvin-diameter of roughly 12 nm. As a consequence the range when the effects of SVOC dimerization start to be important is set to larger diameters. It*

*would be good if you could specifiy why the values were chosen that way. Also, e.g. Kdim lacks any explanation.*

This notification by the referee is very valuable. We had applied, mistakenly, some parameter values from an old "back of an envelope" calculation and forgotten to double-check them. We have updated Table 1 and replotted Figure 11 with surface tension 0.023 N/m, density 1.4 g/cm$^3$ (values from Tröstl et al. 2016) and molecular weight equaling to M/rho (2.15*10$^{-4}$ m$^3$/mol). The value for $K_{dim}$ is taken from Apsokardu and Johnston (2018), who based their value on a study by Ervens and Volkamer (2010). We added all relevant references and explanations to the Table 1. The "tuning" of the figures occurs through choosing ELVOC and SVOC concentrations with which the model results end up in a reasonable magnitude. Promisingly, such ELVOC and SVOC concentration levels are also reasonable for atmospheric conditions.

• *Fig. 4 and Section 3.2. Whenever the authors correlate the GR with a particle size they use the initial size of the growth. Growth rates are inferred from a minimum size to a maximum size, and as GR and the observed growth period varies as the authors point out in Sec. 2.2 I would assume that the mean size of the growth rate measurement gives a more representative value for the diameter where the GR is actually observed.*

We also considered this issue but realized through trying that choosing mean size of the observed growing mode causes artificial bias to the results (see Fig R2 below). This occurs, because, while limiting the minimum duration of the growth periods, the higher growth rates automatically lead to larger mean (and end) diameters for the modes that started at the same initial diameter. By choosing mean or end diameter of the growing modes we would overestimate the impact of diameter on GR.'

We added to the manuscript (Sect 3.2) the following clarification:
"We chose the initial diameter of the growing mode, instead of e.g. the mean diameter, for describing the impact of particle diameter on GR, because applying the mean diameter of the growing mode would cause an artificial bias to the results (if two growth periods with similar duration and different GRs started at same diameter, the one with higher GR would have larger mean diameter than the one with lower GR; this would result in positive correlation between GR and mean diameter, even though the diameters were the same in the beginning and thus the reason for different growth rates should not be the diameter.)"

[Figure]

Figure R2, not added to the manuscript. Observed particle growth rate as a function of initial (left), mean (middle) and end diameter (right) of the growing mode during April-September.

*Technical corrections:*
*• Please consider to cleanup your Figures. Generally I recommend using bigger axis ticks to make the axis better readable. Additionally, while, e.g. Figure 10 has very well readable axis labels, this is not the case for Figures 1-4 and 6.*

We have generally improved the figures, softened the colors and increased the fonts.

*• Please check carefully the usage of definite articles, e.g. p.2 l.8 "by condensation growth", p.2 l.11 "the importance of growth", p2. l.13 "fraction of CCN originating from growth of smaller particles", p.8 l.15 "that we inspected was temperature", p.10 l.27 "50 to 60 nm with temperature, monoterpene concentration", etc.*

Corrected

*• Page 3, lines 6-8. I would point towards Tröstl et al., 2016 (Nature), because they directly describe the Kelvin effect for organics and its influence on growth.*

Done

*• Fig.5, Fig.7 and Fig. 9 I am just wondering, if a reduction of used bins would make the Figures far easier to read and understand, without losing the main conclusions.*

Since the studied variables - temperature, particle size range, and condensation sink - all correlate on some level, as shown in our manuscript, we find it is necessary to limit one of the factors to narrow enough bin in order to study the impact of the others. Thus, we prefer not to reduce the number of bins by making them wider. On the other hand, we also oppose showing e.g. only every second bin, because by showing them all, we demonstrate the consistent behavior of GR as a function of these variables.

*• Page 9, l. 6-15. This paragraphs lacks a conclusion. Monoterpene concentrations are expected to have a weaker correlation than temperature, as temperature not only controls the emissions but also the reaction rates. Given the negative correlation found with the CS and discussed in Sec. 3.2.2. it seems to be logical*
*that the correlation with monoterpene oxidation rate is the strongest. This could be pointed out.*

We are slightly confused with this comment since it seems to point to a paragraph where the commented issues are not discussed. Thus, we break this comment down to pieces and respond to them separately.

*The lack of conclusion of the pointed paragraph.* We think there was a conclusion but agree that the explanation was not clear enough for making it easy to understand. We rearranged the last sentences of the paragraph (page 10, lines 22-27) for clarifying the conclusion.
*Impact of temperature on monoterpene concentrations via temperature dependency of the reaction rates.* We estimate that the temperature dependency of the reaction rates is negligible in comparison to the temperature dependency of the emissions. Where the increase of temperature from 270 K to 300 K increases the reaction rate between MT and $O_3$ by a factor of 1.2 ($k_{OH+MT}$ = a

*exp(-580/T), from Atkinson et al., 2006), the emission rate increases by a factor of 27 (E = a*exp(0.10*(T-303.15)), Tarvainen et al., 2005).

*Correlation between CS and GR.* We agree that due to the positive correlation (probably misspelled by the referee as negative correlation) between CS and GR it is logical that monoterpene oxidation rate correlates better with GR than calculated oxidation product concentration. However, since we inspect separately the relation between CS and GR only in the next Section, we prefer not to discuss it here in order not to confuse the reader.

• *Page 12, l. 13 and Fig. 11 a. While in the text Ke is set to 1 the Figure legend says Ke=0.*

The figure legend is corrected to Ke = 1.

Atkinson, R., Baulch, D. L., Cox, R. A., Crowley, J. N., Hampson, R. F., Hynes, R. G., Jenkin, M. E., Rossi, M. J., Troe, J., and IUPAC Subcommittee: Evaluated kinetic and photochemical data for atmospheric chemistry: Volume II – gas phase reactions of organic species, Atmos. Chem. Phys., 6, 3625–4055, doi:10.5194/acp-6-3625-2006, 2006.

Tarvainen, V., Hakola, H., Hellén, H., Bäck, J., Hari, P., and Kulmala, M.: Temperature and light dependence of the VOC emissions of Scots pine, Atmos. Chem. Phys., 5, 989–998, doi:10.5194/acp-5-989-2005, 2005.

[revised manuscript text omitted]